# The gut microbiota modulates host amino acid and glutathione metabolism in mice

Adil Mardinoglu[1,2,†,*], Saeed Shoaie[1,†], Mattias Bergentall[3,4], Pouyan Ghaffari[1], Cheng Zhang[2], Erik Larsson[3,4], Fredrik Bäckhed[3,4] & Jens Nielsen[1,2]

## Abstract

**The gut microbiota has been proposed as an environmental factor that promotes the progression of metabolic diseases. Here, we investigated how the gut microbiota modulates the global metabolic differences in duodenum, jejunum, ileum, colon, liver, and two white adipose tissue depots obtained from conventionally raised (CONV-R) and germ-free (GF) mice using gene expression data and tissue-specific genome-scale metabolic models (GEMs). We created a generic mouse metabolic reaction (MMR) GEM, reconstructed 28 tissue-specific GEMs based on proteomics data, and manually curated GEMs for small intestine, colon, liver, and adipose tissues. We used these functional models to determine the global metabolic differences between CONV-R and GF mice. Based on gene expression data, we found that the gut microbiota affects the host amino acid (AA) metabolism, which leads to modifications in glutathione metabolism. To validate our predictions, we measured the level of AAs and N-acetylated AAs in the hepatic portal vein of CONV-R and GF mice. Finally, we simulated the metabolic differences between the small intestine of the CONV-R and GF mice accounting for the content of the diet and relative gene expression differences. Our analyses revealed that the gut microbiota influences host amino acid and glutathione metabolism in mice.**

**Keywords** genome-scale metabolic models; germ-free mice; glutathione metabolism; metabolomics; transcriptomics
**Subject Categories** Genome-Scale & Integrative Biology; Metabolism; Microbiology, Virology & Host Pathogen Interaction
**Mol Syst Biol. (2015) 11: 834**

## Introduction

The human gut harbors a vast ensemble of bacteria that have profound effects on host physiology (Huttenhower *et al*, 2012).

Complex disorders including obesity (Ley *et al*, 2006; Turnbaugh *et al*, 2009), type 2 diabetes (T2D) (Qin *et al*, 2012; Karlsson *et al*, 2013), atherosclerosis (Wang *et al*, 2011b; Karlsson *et al*, 2012), and non-alcoholic fatty liver disease (NAFLD) (Henao-Mejia *et al*, 2012) as well as the opposite end of the spectrum, for example, malnutrition (Smith *et al*, 2013; Subramanian *et al*, 2014), have been associated with dysbiosis in the human gut microbiota. To gain mechanistic insights into the contribution of specific microbial populations to the progression of such disorders, germ-free (GF) animals (e.g. mice and rats) have been adopted for studying the association of the gut microbiota with disease pathogenesis (Ridaura *et al*, 2013).

Comparisons between GF and conventionally raised (CONV-R) mice are often used for studying the effect of gut microbiota on host physiology (Wostmann, 1981; Stappenbeck *et al*, 2002; Claus *et al*, 2008; Slack *et al*, 2009; El Aidy *et al*, 2013). Moreover, Larsson *et al* (2012) studied the response of the host induced by microbiota along the length of the gut in CONV-R and GF C57Bl6/J mice and provided a detailed description for tissue-specific host transcriptional responses.

Global metabolic differences of cells/tissues between different clinical conditions can be revealed through the use of genome-scale metabolic models (GEMs) (Mardinoglu & Nielsen, 2012, 2015; Yizhak *et al*, 2013, 2014a,b; Bordbar *et al*, 2014; Shoaie & Nielsen, 2014; O'Brien *et al*, 2015; Varemo *et al*, 2015; Zhang *et al*, 2015). GEMs include the known metabolism-related reactions and associated genes in a particular cell and tissue and serve as an excellent scaffold for the integration of omics data (e.g. proteomics, transcriptomics, and metabolomics) for increasing our understanding of the relationship between genotype and phenotype (Mardinoglu *et al*, 2013b). To date, simulation-ready cell-/tissue-specific GEMs (Gille *et al*, 2010; Karlstaedt *et al*, 2012; Mardinoglu *et al*, 2013a, 2014a) and automatically reconstructed GEMs (Jerby *et al*, 2010; Agren *et al*, 2012; Wang *et al*, 2012; Yizhak *et al*, 2014a; Uhlen *et al*, 2015) have been used for studying the metabolism of cells/tissues in health and disease states.

In order to examine the gut microbiota-induced transcriptional responses of the host metabolism, we performed microarray

1   Department of Biology and Biological Engineering, Chalmers University of Technology, Gothenburg, Sweden
2   Science for Life Laboratory, KTH – Royal Institute of Technology, Stockholm, Sweden
3   Department of Molecular and Clinical Medicine, Wallenberg Laboratory, University of Gothenburg, Gothenburg, Sweden
4   Novo Nordisk Foundation Center for Basic Metabolic Research, Section for Metabolic Receptology and Enteroendocrinology, Faculty of Health Sciences, University of Copenhagen, Copenhagen, Denmark
    *Corresponding author. Tel: +46 31 772 3140; Fax: +46 31 772 3801; E-mail: adilm@scilifelab.se
    †These authors contributed equally to this work

analysis of liver as well as epididymal and subcutaneous white adipose tissues (WATs) obtained from both CONV-R and GF mice, and analyzed the global gene expression profile of these tissues together with the previously published gene expression profiles of duodenum, jejunum, ileum, and colon tissues. We created a generic mouse metabolic reaction (MMR) GEM and generated tissue-specific mouse GEMs primarily based on proteomics data. We investigated the metabolic differences between CONV-R and GF mice using global gene expression profiling of the host tissues and the network topology provided by the tissue GEMs, and validated our predictions by generating metabolomics data for these two sets of mice. Finally, we revealed the metabolic differences between the small intestine of CONV-R and GF mice accounting for the content of the chow diet as well as the relative gene expression differences using relative metabolic differences (RMetD) method.

## Results

### Global transcriptional profiles of CONV-R and GF mice

CONV-R and GF C57Bl6/J male mice were fed autoclaved chow diet *ad libitum* and then euthanized at 12–14 weeks of age (Larsson *et al*, 2012). We isolated RNA from liver as well as epididymal and subcutaneous WATs obtained from CONV-R and GF mice and performed global transcriptome analysis (Fig 1A). Small intestine and colon have been previously removed from the same two sets of mice, and small intestine has been divided into eight whereas the colon into three equal-sized segments (Larsson *et al*, 2012). Global transcriptome analysis was performed for the first (duodenum), fifth (jejunum), and eighth (ileum) segments of the small intestine and the proximal piece of the colon (Fig 1A).

We performed principal component analysis (PCA) of the transcription profiles for the tissues separately and observed a clear separation between the CONV-R and GF mice for duodenum, jejunum, ileum, colon, and liver, whereas no separation was found for both WATs (Fig EV1). We identified significantly differentially expressed probe sets and genes in MMR, from here on referred as metabolic genes, by comparing gene expression profiles of tissues obtained from CONV-R versus GF mice (Fig 1B, Dataset EV1). During the identification of the significantly ($Q$-value < 0.05) differentially expressed probe sets and metabolic genes, we adjusted $P$-values using the false discovery rate (FDR) method and calculated $Q$-values. We found that ileum tissue had the largest number of differentially expressed metabolic genes between CONV-R and GF mice, and it was followed by duodenum, jejunum, colon, and liver tissues (Fig 1B). It should also be noted that we only detected two significantly differentially expressed metabolic genes between the subcutaneous WAT whereas no differentially expressed metabolic genes between the epididymal WAT.

Comparing the differentially expressed metabolic genes between duodenum, jejunum, ileum, colon, and liver tissues of CONV-R and GF mice (Fig 1C), we found that the expression of the nicotinamide nucleotide transhydrogenase (Nnt) gene is higher and ectonucleoside triphosphate diphosphohydrolase 4 (Entpd4) is lower in all five tissues of CONV-R mice compared with GF mice

(Fig 1D). Strikingly, we found that Nnt and Entpd4 are also the only differentially expressed genes in the subcutaneous WAT of the CONV-R mice compared with GF mice and followed the same directional changes in the subcutaneous WAT as in all other five analyzed tissues. Entpd4 was initially named human Golgi UDPase, and it hydrolyzes nucleoside diphosphates. UDP is the best substrate for this enzyme, and its ADP activity is insignificant. Nnt is required for regular mitochondrial function, and it uses energy from the mitochondrial proton gradient to transfer reducing equivalents from NADH to NADPH. The resulting NADPH is used for driving macromolecular biosynthesis as well as for the reduction of glutathione (GSH) (Fig 1D). Here, we focused on the metabolic differences associated with Nnt due to its well-known metabolic function.

### Creation of MMR and reconstruction of mouse tissue-specific GEMs

We constructed MMR by using the mouse orthologs of human genes in HMR2 (Mardinoglu *et al*, 2014a) (Fig 2A), and the resulting generic model includes 8,140 metabolism-related reactions, 3,579 associated metabolic genes to those reactions, and 5,992 metabolites in eight different subcellular compartments. Previously, stable isotope labeling with amino acids (SILAC)-based proteomics was generated to analyze the expression of 7,349 proteins in 28 different major C57BL/6 mouse tissues (Geiger *et al*, 2013) and these data cover 2,030 of the protein-coding genes in MMR (Fig 2B, Dataset EV2). We reconstructed tissue-specific GEMs for 28 mouse tissues by using proteomics data, MMR, and the tINIT algorithm (Agren *et al*, 2014) (see Materials and Methods). The tINIT algorithm allows for the reconstruction of functional GEMs based on global proteomics data as well as user-defined metabolic tasks, which the resulting model should be able to perform. During the reconstruction of the models, we complemented 56 metabolic tasks (functions) (Agren *et al*, 2014), which are known to occur in all cells/tissues.

The number of reactions, metabolites, and genes incorporated in the models are presented in Dataset EV3. A total of 5,813 reactions, 4,574 metabolites, and 1,838 genes were shared across the tissue-specific GEMs of which 2,750 (47.3%) reactions, 3,001 (65.6%) metabolites, and 669 (36.4%) genes were common to all tissue-specific GEMs. We found that 322 reactions, 134 metabolites, and 120 genes were incorporated into only one specific GEM (Fig 2C). By pairwise comparison of GEMs, we found that each model has an average of 765 reactions (Dataset EV4), 430 metabolites (Dataset EV5), and 342 genes (Dataset EV6) different from other tissues where the muscle tissue was the one with the highest average difference (Fig 2C).

We analyzed the heterogeneity of the mouse tissue-specific GEMs (Fig 2D) in terms of incorporated reactions, genes, and metabolites by calculating the heterogeneity degree of each model. The heterogeneity degree allowed us to capture the divergence between metabolic networks based on their constituent parameters including reactions, metabolites, and genes, and it was calculated using the average and maximum Hamming distance of the models (Ghaffari *et al*, 2015). Moreover, we analyzed the heterogeneity of recently reconstructed human cell-specific GEMs (Agren *et al*, 2014) that have been reconstructed based on antibody-based proteomics data

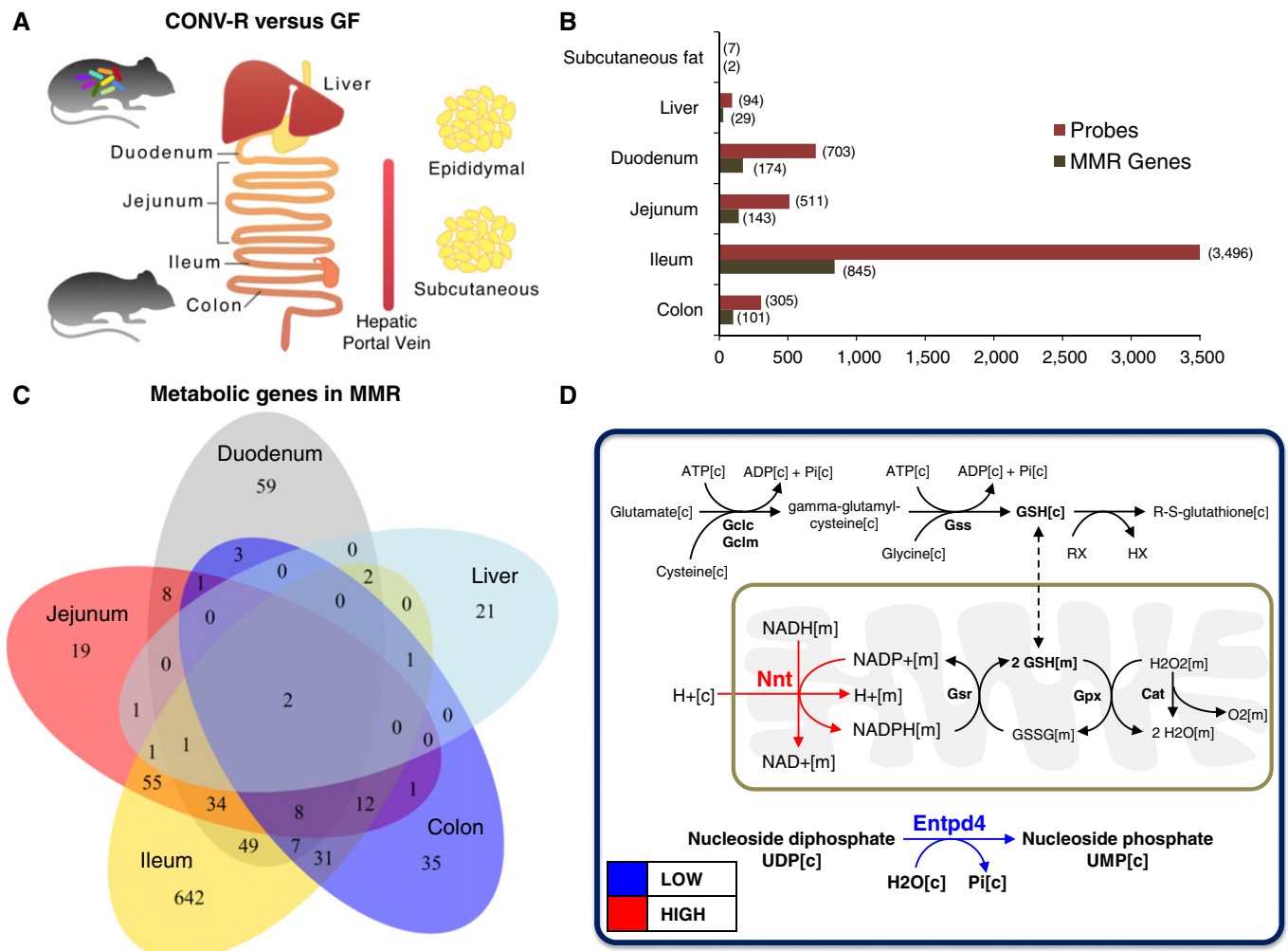

**Figure 1.  Global gene expression profiling of tissues obtained from CONV-R and GF mice.**

A  Liver as well as epididymal and subcutaneous WATs was obtained from both CONV-R and GF mice, and global gene expression profiling was generated using microarrays. Transcriptomics data for these three tissues were analyzed together with the previously published gene expression profiling of duodenum, jejunum, ileum, and colon tissues.

B  Gene expression data for each tissue were normalized independently of other tissues, and significantly (Q-value < 0.05) differentially expressed probe sets and metabolic genes in Mouse Metabolic Reaction database were presented in each analyzed tissue.

C  The overlap between the significantly (Q-value < 0.05) and differentially expressed metabolic genes in duodenum, jejunum, ileum, colon, and liver is presented.

D  The significantly (Q-value < 0.05) and differentially expressed metabolic genes, Nnt, and Entpd4, as well as the reactions associated with Nnt, are presented. Red and blue arrows indicate the significantly higher (Q-value < 0.05) and lower expression of the metabolic genes in CONV-R mice compared to GF mice, respectively.

in the Human Protein Atlas (www.proteinatlas.org) (Uhlen *et al*, 2010, 2015; Kampf *et al*, 2014b). On average, mouse tissue-specific GEMs showed an average heterogeneity degree of 0.77 for reactions, 0.72 for metabolites, and 0.78 for genes, whereas human cell-specific GEMs had an average heterogeneity degree of 0.8 for reactions, 0.7 for metabolites, and 0.84 for genes (Fig 2D). Compared with the human cell-specific GEMs, the mouse tissue-specific GEMs had a slightly higher metabolic uniformity and lower heterogeneity based on the incorporated genes and reactions, but they had a slightly higher heterogeneity based on the incorporated metabolites into the models.

We next incorporated the significantly differentially expressed genes between CONV-R and GF mouse tissues and generated four functional GEMs for liver (*iMouseLiver*), adipose (*iMouseAdipose*), colon (*iMouseColon*), and small intestine (*iMouseSmallintestine*), with the latter reconstructed by merging the GEMs for duodenum, jejunum, and ileum tissues. During manual evaluation of the GEMs, previously published functional human cell-type GEMs for hepato-cytes in liver tissue (Mardinoglu *et al*, 2014a) and adipocytes in adipose tissue (Mardinoglu *et al*, 2013a, 2014b) were also used to include known biological functions to the GEMs. The number of the incorporated reactions, metabolites, and genes in the four functional annotated tissue-specific GEMs as well as in the draft GEMs is provided in Dataset EV3. MMR as well as the all mouse tissue-specific models are publicly available in systems biology markup language (SBML) format at the Human Metabolic Atlas portal (www.metabolicatlas.org) (Pornputtapong *et al*, 2015), at the BioModels database and as Computer Code EV1.

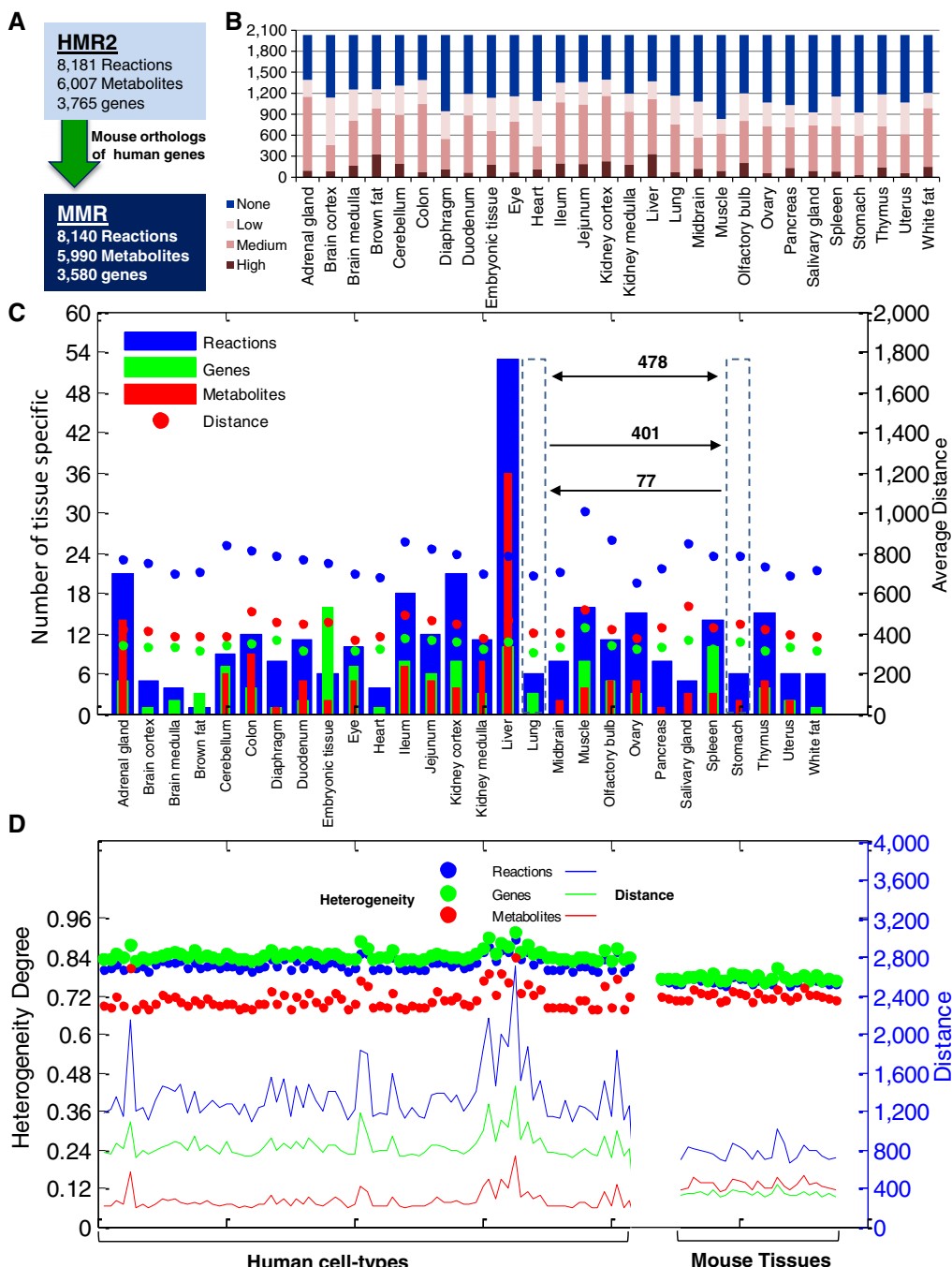

**Figure 2.   Creation of MMR and generation of tissue-specific GEMs.**

A   Mouse Metabolic Reaction database (MMR) was created using the mouse orthologs of human genes based on Human Metabolic Reaction database 2.0 (HMR2).

B   The expression level of the 2,032 proteins used in the generation of the 28 tissue-specific mice models is presented.

C   Bar plots represent the distribution of tissue-specific reactions, metabolites, genes, and metabolites across the 28 mouse tissue GEMs. Filled circles depict average distance of each tissue GEMs compared to others. Average distance, calculated based on Hamming distance method, indicates required alteration to transform one tissue model to the other based on the reactions and metabolites and genes. For instance, 478 changes in gene profile are required for intertransformation of GEM for lung and stomach, from which 401 changes in genes correspond to transformation of lung to stomach and 77 changes in genes correspond to transformation of stomach to lung.

D   Filled circles represent the heterogeneity degree of 28 mouse tissues and 83 healthy human cell types. Heterogeneity values are projected on the left hand side axis. There is a fall, ~0.06 degree, in heterogeneity of mice models compared to human modes based on genes and a lower decrease, ~0.03, based on the reactions. However, comparing mouse tissues to human cells revealed higher heterogeneity based on the metabolites, in contrast to the reaction and genes. Average Hamming distance of GEMs for mouse tissues and human cell types are projected on the right hand axis. Mouse tissues have relatively less, 40–60%, inter-model distance compared to human cells based on the reactions and genes. However, the trend is reversed with around 50% increased inter-model distance for metabolites. In general, mouse tissue-specific GEMs show gain of heterogeneity based on metabolites and loss of heterogeneity based on genes.

### Decreased glutathione synthesis in the small intestine of CONV-R mice

We compared the gene expression profiling in the small intestine segments (duodenum, jejunum and ileum) of CONV-R and GF mice, and examined the changes in the expression of the genes interacting with Nnt using the network structure provided by *iMouse-Smallintestine* (Fig 1D). We found that the expression of glutathione reductase (Gsr) which uses NADPH as an electron donor to reduce glutathione disulfide (GSSG) to GSH was also significantly higher (*Q*-value < 0.05) in all three small intestine segments of CONV-R mice compared to GF mice (Fig 3A, Dataset EV1). GSH plays a key role in reducing oxidative stress, and it can be synthesized within the cells from glutamate, cysteine, and glycine through

the use of glutamate-cysteine ligase catalytic subunit (Gclc), glutamate-cysteine ligase modifier subunit (Gclm), and glutathione synthetase (Gss). We found that the expression of Gclc is significantly lower (*Q*-value < 0.05) in jejunum and ileum, and the expressions of Gclm and Gss are significantly lower (*Q*-value < 0.05) in the ileum of CONV-R mice compared to GF mice (Fig 3A). Based on gene expression data, we observed that decreased *de novo* synthesis of GSH in the small intestine of CONV-R mice was compensated by higher expression of Nnt and Gsr compared to GF mice.

The decreased synthesis of the GSH in the small intestine segments of CONV-R mice may be due to the limited availability of the substrates including glutamate, cysteine, and glycine. Hence, we examined the expression of the enzymes involved in the synthesis and catabolism of these amino acids by differentially expressed

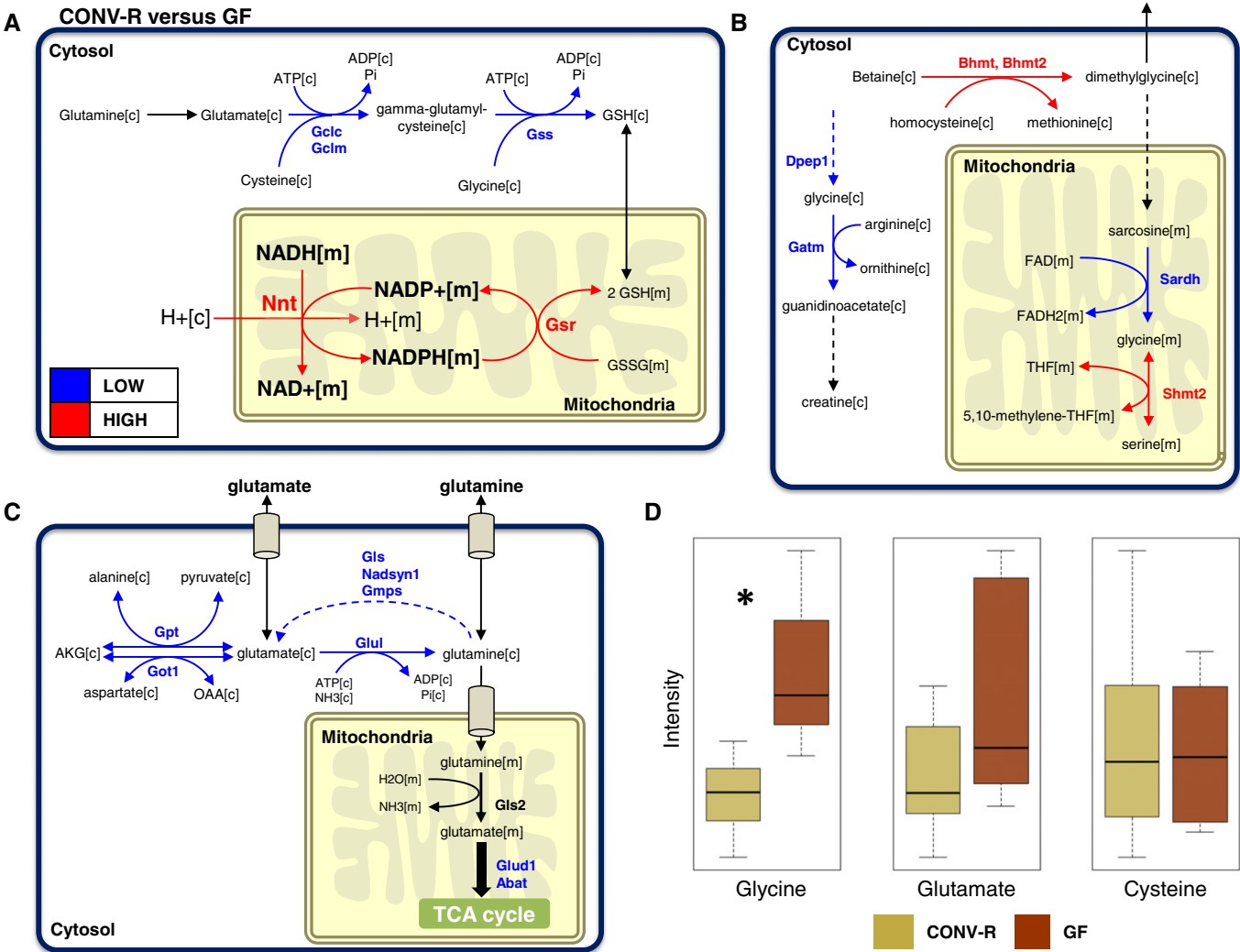

**Figure 3. Metabolic differences in the small intestine.**

A Metabolic genes as well as the associated reactions involved in the formation of glutathione (GSH) are presented.

B, C Significant differences associated with (B) glycine and (C) glutamine are shown. Red and blue arrows indicate the significantly (*Q*-value < 0.05) higher and lower expression of the metabolic genes in CONV-R mice compared to GF mice, respectively.

D The levels of glycine, glutamine, and cysteine used in the *de novo* synthesis of the GSH are measured in the hepatic portal vein that conducts blood from the gastrointestinal tract to the liver tissue. *Q*-value < 0.05.

genes in the small intestine (Dataset EV7) and using the network topology provided by *iMouseSmallintestine*. We integrated differentially expressed genes in duodenum, jejunum, and ileum using the lowest *Q*-value for the genes and associated fold changes for studying the metabolic differences between the small intestine of CONV-R and GF mice (Dataset EV7). Hereby, we identified significantly (*Q*-value < 0.05) differentially expressed genes linked to biosynthesis of glycine (Fig 3B) and glutamate (Fig 3C) and found that there are metabolic differences in the utilization of these AAs between CONV-R and GF mice. In contrast, we did not detect any significant change in the expression of the genes linked to cysteine except Gclc and Gclm (Dataset EV7).

In healthy animals, AAs in the small intestine are released in the plasma and utilized by other peripheral tissues (e.g. liver, adipose, and muscle tissues). Considering the down-regulation of genes associated with *de novo* synthesis of GSH and of glutamate and glycine required for GSH biosynthesis in the small intestine in CONV-R mice, we hypothesized that the plasma level of glutamate and glycine secreted from the small intestine in CONV-R mice may also be lower compared with GF mice. Hence, we measured the level of these two AAs in the hepatic portal vein (PV), which conducts blood from the gastrointestinal tract to the liver, of CONV-R and GF mice, and found that the PV level of glycine was significantly lower (ANOVA test, *Q*-value < 0.05) and glutamate is slightly lower in CONV-R mice compared to GF mice (Fig 3D, Dataset EV8). We also measured the level of cysteine, likewise used for biosynthesis of GSH, in the PV of CONV-R and GF mice (Fig 3D) but found no differences, in agreement with the fact that there were no significant changes in the expression of genes encoding enzymes required for cysteine utilization (Dataset EV7).

We also searched for global metabolic differences in the small intestine by mapping the gene expression data (Dataset EV7) to the network topology provided by the small intestine GEM using the reporter subnetworks algorithm (Patil & Nielsen, 2005). Hereby, we found that there are major metabolic differences around 17 other AAs (Dataset EV9). We thus next measured the level of these 17 AAs in the PV and found that the levels of arginine, asparagine, histidine, isoleucine, leucine, methionine, phenylalanine, proline, serine, threonine, tryptophan, tyrosine, and valine were significantly lower and the level of glutamine was significantly higher in CONV-R mice compared to GF mice. We did not detect any significant changes in the level of alanine, aspartate, and lysine (Fig 4A, Dataset EV8). Our results indicate that the gut microbiota alter AA metabolism of the host.

## Metabolic differences between the liver tissues of CONV-R and GF mice

We found that the expression of Nnt was significantly higher and Entpd4 was significantly lower in the liver tissue of CONV-R compared to GF mice (Fig 5A) and validated the expression of these genes by quantitative reverse transcription PCR (RT–PCR) methods (Fig 5B). We found that glutathione S-transferase pi 1 (Gstp1), which has a role in GSH metabolism, metabolism of xenobiotics by cytochrome P450, and drug metabolism, is significantly higher in CONV-R mice compared to GF mice. Notably, Claus *et al* (2008) measured the liver tissue level of GSSG, which is used as a substrate for the reaction catalyzed by the Gsr in CONV-R and GF mice by employing a high-resolution $^1$H NMR spectroscopic approach, and reported that the liver tissue level of GSSG was significantly higher in CONV-R mice. Hence, we hypothesized that higher Nnt expression in CONV-R mice might be the response of liver to the lower level of glycine required for the *de novo* synthesis of the GSH. Strikingly, Claus *et al* (2008) has also measured the glycine level in the liver tissue of CONV-R and GF mice, and found that the level of glycine is lower in CONV-R mice compared to GF mice. We also measured the PV level of serine, which can be taken up by the liver and converted to glycine, and found that the level of serine was also significantly lower in CONV-R mice compared to GF mice (Fig 4A).

However, N-acetylated AAs can also be taken up by the liver and hydrolyzed to acetate and a free AA by aminoacylases, ACY1, ACY2 (ASPA), and ACY3 (Fig 4B) which are predicted to be present in human liver tissue based on the transcriptomics and proteomics data (Kampf *et al*, 2014a; Uhlen *et al*, 2015). Accordingly, we measured the level of the acetyl-AAs in the PV and found that the levels of N-acetyllysine, N-acetylalanine, N-acetylarginine, N-acetyl-cysteine, N-acetylglutamate, N-acetylglycine, N-acetylisoleucine, N-acetylleucine, N-acetylmethionine, N-acetylphenylalanine, N-acetylthreonine, N-acetyltryptophan, N-acetyltyrosine, and N-acetyl-valine are significantly lower (*Q*-value < 0.05), whereas N-acetylser-ine and N-acetylthreonine are slightly (0.05 < *Q*-value < 0.1) lower in CONV-R mice compared to GF mice (Fig 4C, Dataset EV8). We could not detect any significant changes in the level of N-acetyl-asparagine, N-acetylaspartate, N-acetylglutamine, and N-acetyl-histidine.

Moreover, we investigated the differences in the expression of liver tissue genes between CONV-R and GF mice. We found that the expression of Alas1 involved in glycine metabolism, Slc2a1 known as Glut1 and involved in glucose transportation, Slc16a12 involved in creatine transportation, and Hmgcr involved in cholesterol synthesis as well as Cyp7a1 and Akr1d1 involved in bile acid synthesis were significantly lower in the liver tissue of CONV-R mice compared to GF mice (Fig 5A). On the other hand, the expressions of Sdr9c7 involved in vitamin A metabolism, Elovl3 involved in long-chain fatty acids (FAs) elongation cycle, and Uap1l1 involved in amino sugar and nucleotide sugar metabolism were significantly upregulated in liver tissue of CONV-R mice compared to GF mice.

## Metabolic differences between the colon tissues

We found that the expression of Sardh that converts sarcosine to the glycine in the mitochondria is significantly lower (*Q*-value < 0.05), whereas the expression of the genes involved in GSH metabolism including, Gsta4, Gstk1, Gstp1, and Gstt1 is significantly higher (*Q*-value < 0.05) in the colon tissue of CONV-R mice compared to GF mice (Fig 6). Even though degradation of betaine is lower, the increased expression of glutathione transferases (Gsts) may be explained by the increased expression of the Nnt in the colon tissue of CONV-R mice similar to the liver and small intestine tissues of CONV-R mice.

We also revealed the global metabolic differences (Dataset EV1) between the colon tissue of CONV-R and GF mice by mapping the significantly differentially expressed genes to *iMouseColon* (Fig 6). We found that the expression of Arg2 involved in arginine metabolism as well as Ces1g, Aldh1a2, and Rbp4 involved in vitamin A

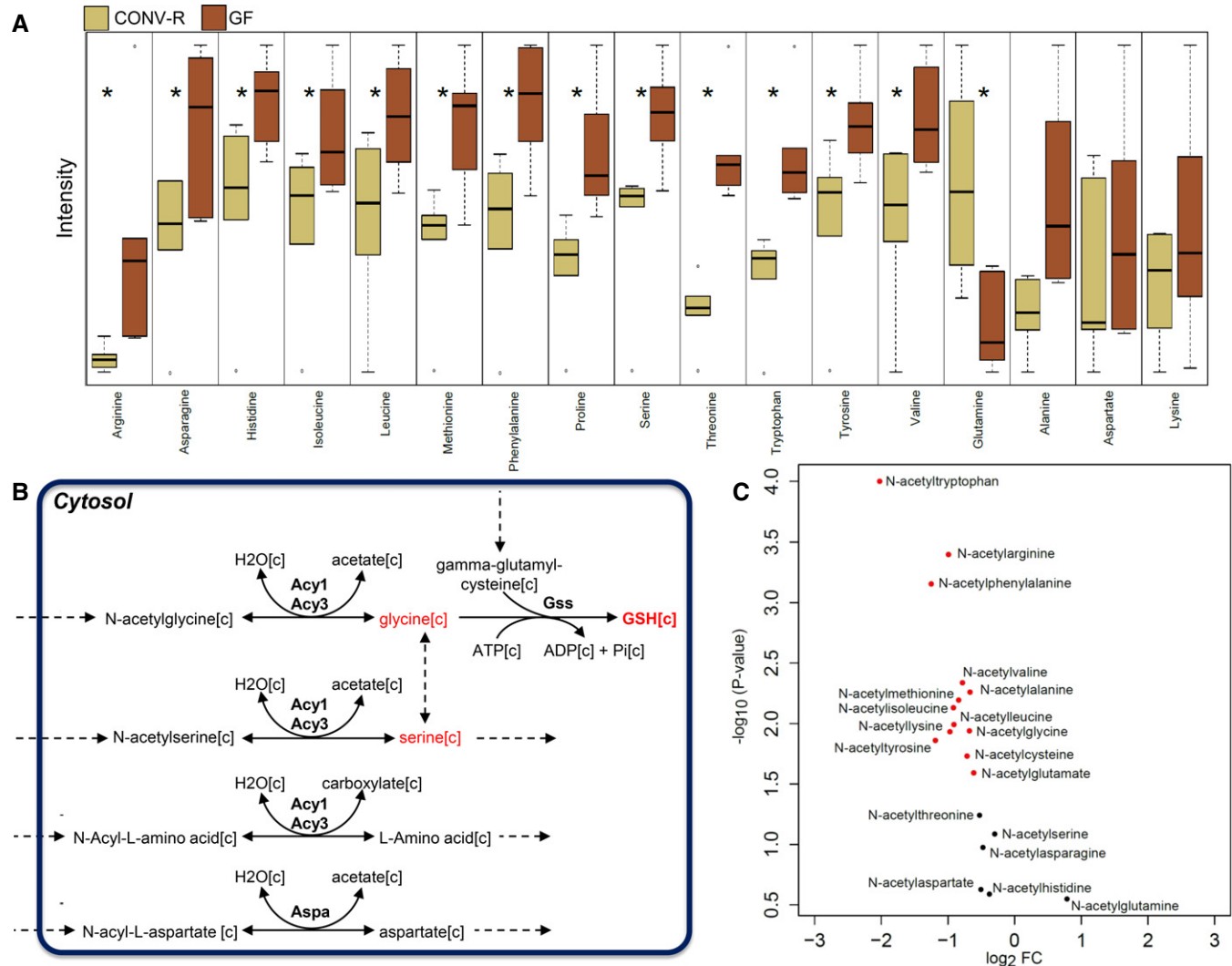

**Figure 4. The level of the AAs and N-acetylated AAs in the hepatic portal vein.**

A  The level of the significantly (*Q*-value < 0.05) changed amino acids (AAs) including arginine, asparagine, glutamine, histidine, isoleucine, leucine, methionine, phenylalanine, proline, serine, threonine, tryptophan, tyrosine, valine, and glutamine as well as the non-significantly changed AAs including alanine, aspartate, and lysine are measured in the hepatic portal vein of CONV-R and GF mice.

B  Reactions involved in the hydrolysis of N-acetylated AAs to acetate and a free AA as well as their catalyzing enzymes, aminoacylases, ACY1, ACY2 (ASPA), and ACY3 are presented.

C  The level of the N-acetylated AAs in hepatic portal vein of CONV-R and GF mice are shown in the volcano plot.

metabolism was higher in the colon tissue of CONV-R mice compared to GF mice. On the other hand, we found that Aldob and Aldh9a1 involved in glycolysis; Hmgcs1, Hsd17b7, Nsdhl, and Sc4 mol involved in cholesterol synthesis; Mgam and Sis involved in starch and sucrose metabolism; and Slc2a5, Slc2a9, Sord, and Khk involved in fructose metabolism as well as Ace2 transcription factor involved in the conversion of angiotensin were significantly lower in CONV-R mice (Fig 6). Moreover, we found that the expression of genes involved in the transport of AAs is significantly lower in the colon tissue of CONV-R compared with GF mice (Dataset EV1). Our analysis indicated that the overall central metabolism of the colon tissue may have reduced activity in CONV-R mice compared to GF mice based on the gene expression data.

## The interactions between the microbiota in the small intestine

When CONV-R and GF C57Bl6/J mice were fed with a standard autoclaved chow diet, we found that CONV-R mice were eating approximately 20% more than the GF mice. However, the PV level of 14 AAs and 13 N-acetylated AAs was significantly lower in CONV-R mice compared to GF mice (Fig 4A and C), and this may be due to the consumption of these AAs by the bacteria in the small intestine.

Bacteroidetes and Clostridium cluster XIVa are among the dominant phyla in the ileum (Zoetendal *et al*, 2012; Van den Abbeele *et al*, 2013). To understand the metabolic interactions between the gut microbiota as well as their interactions with the small intestine

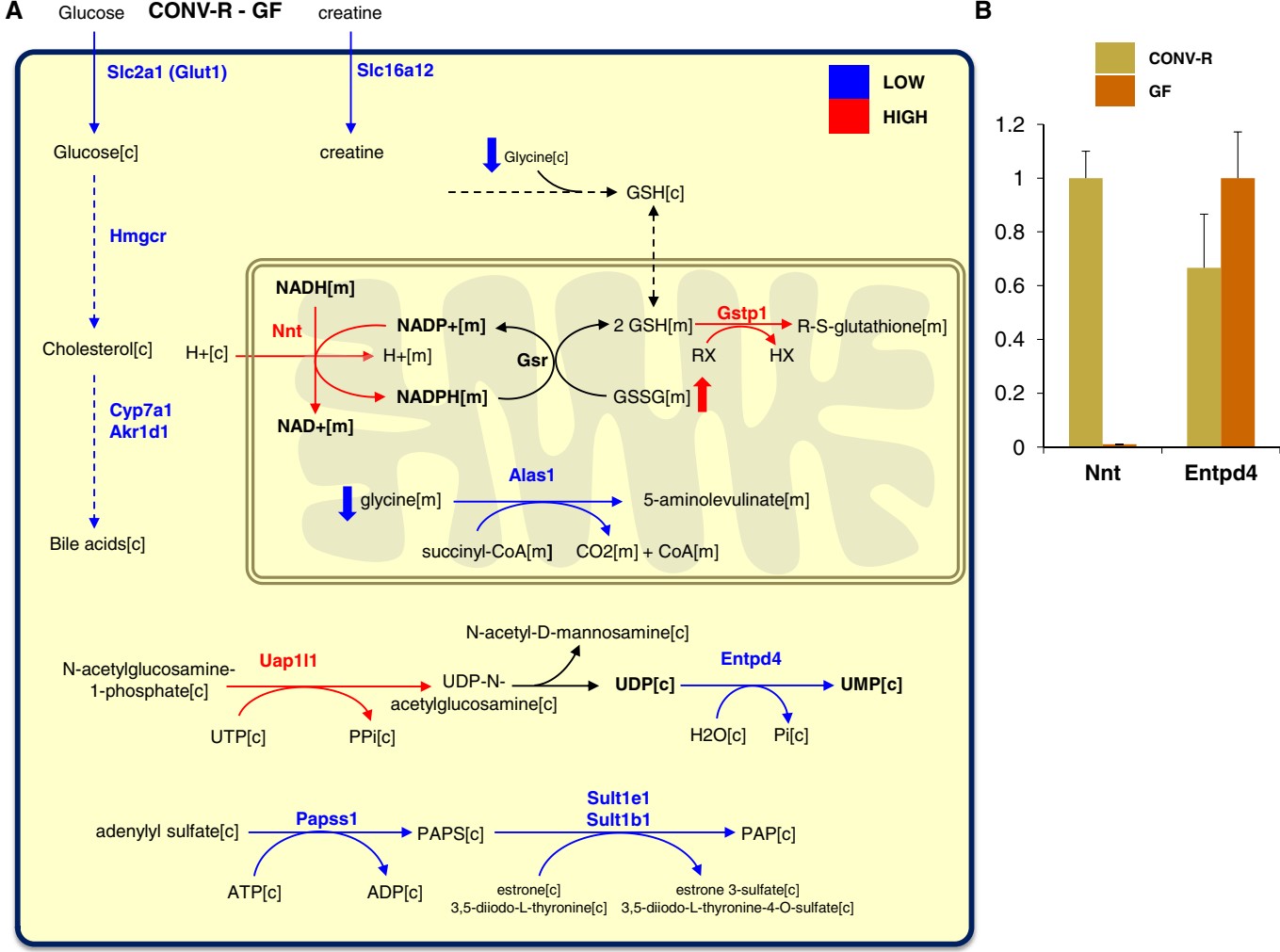

**Figure 5.  Metabolic differences in the liver tissue.**

A    The significantly (Q-value < 0.05) and differentially expressed metabolic genes in liver tissue of CONV-R mice compared to GF mice are mapped to the functional GEM for mice liver tissue. Red and blue arrows indicate the significantly (Q-value < 0.05) higher and lower expression of the metabolic genes in CONV-R mice compared to GF mice, respectively. Thick arrows indicate the liver tissue level of the metabolites.

B    The liver tissue expression of Nnt and Entpd4 was measured by RT–PCR.

in CONV-R mice, we simulated the interplay between the two key species *Bacteroides thetaiotamicron* and *Eubacterium rectale* as relevant representatives of these two main phyla in the human ileum. The Community and Systems-level Interactive Optimization toolbox (CASINO) has recently been developed for studying the interactions between the gut microbiota and comprises an optimization algorithm integrated with diet analysis to predict phenotypes (Shoaie *et al*, 2015). Here, we employed manually reconstructed GEMs for *B. thetaiotamicron* and *E. rectale* (Shoaie *et al*, 2013) to study the interactions between the gut microbiota by using CASINO. During the simulations, we used the content of the autoclaved chow diet (Dataset EV10) and assumed that maximum 40% of the total protein may be consumed by the bacteria (set as upper bound), 5% of the total protein is transferred to the colon tissue, and the remaining proteins are consumed by the small intestine (Fig 7A) based on a previous study where the total AAs in the gastrointestinal tract of CONV-R and GF mice were measured (Whitt & Demoss, 1975). We

also assumed that 40% of the digestible and 5% of the non-digestible carbohydrates in the diet were consumed by the bacteria in the small intestine and 5% of the digestible carbohydrates was transferred to the colon tissue (Gibson & Roberfroid, 1995).

We maximized for the growth of bacteria and predicted the amount of the short-chain fatty acids (SCFAs) (acetate, propionate and butyrate) produced by the *B. thetaiotamicron* and *E. rectale* (Fig 7B and Dataset EV11) by setting the content of the diet as upper bound to the GEMs for bacteria (Dataset EV10). We observed that part of the acetate produced by the *B. thetaiotamicron* is consumed by the *E. rectale* and contributes to the production of butyrate in *E. rectale*. We found that isoleucine, proline, and valine are only consumed by the *E. rectale,* and glycine, serine, alanine, cystine, glutamate, histidine, leucine, lysine, methionine, phenylalanine, threonine, and tyrosine are consumed by both *B. thetaiotamicron* and *E. rectale*, whereas arginine, aspartate, and tryptophan are not consumed by neither of these bacteria (Fig 7C and Dataset EV11). It

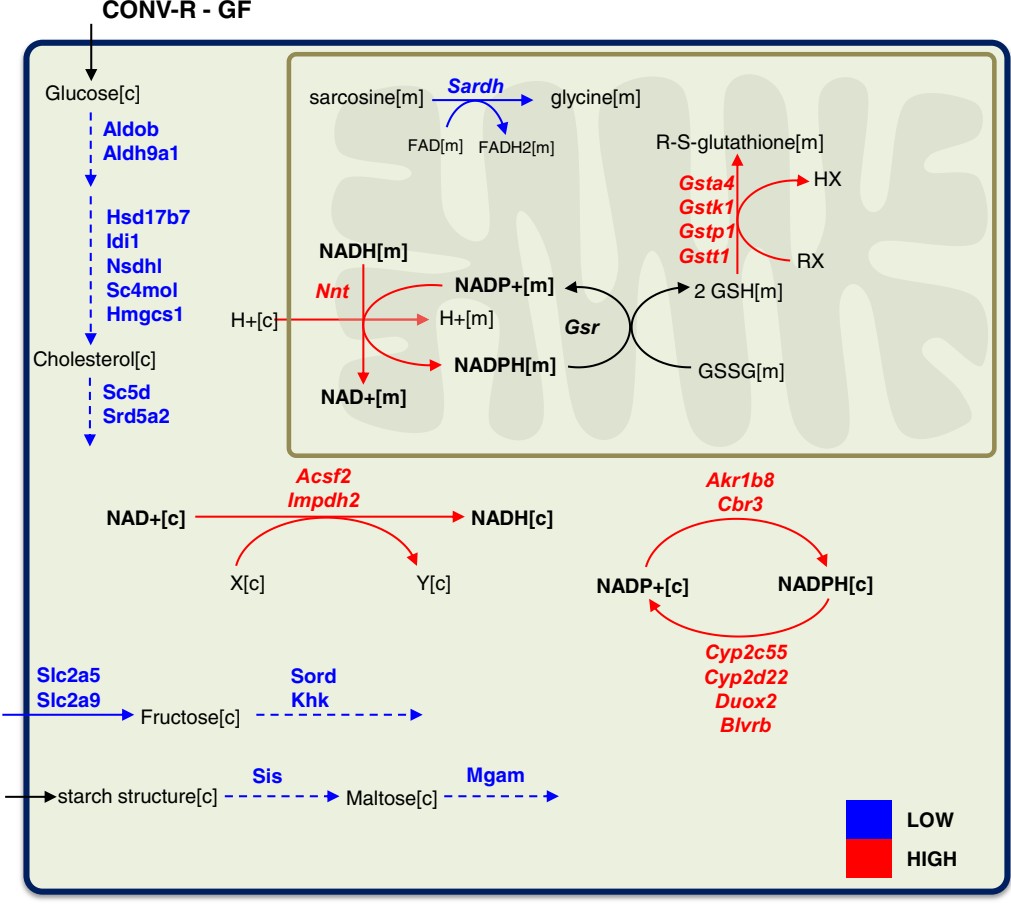

**Figure 6.  Metabolic differences in the colon tissue.**
The significantly (*Q*-value < 0.05) differentially expressed metabolic genes in colon tissue of CONV-R mice compared to GF mice are mapped to the functional GEM for mice colon tissue. Red and blue arrows indicate the significantly (*Q*-value < 0.05) higher and lower expression of the metabolic genes in CONV-R mice compared to GF mice, respectively.

should also be noted that the gut microbiota consumed all of the available glycine, serine, and threonine during the formation of its biomass.

### Metabolic differences between the small intestine of CONV-R and GF mice

We simulated the metabolic differences between the small intestine of CONV-R and GF mice accounting for the relative differences in the global gene expression data of the small intestine using RMetD method (see Materials and Methods). RMetD is developed to integrate the up- and down-regulation of the enzymes together with the calculated corresponding *P*-values into the GEMs rather than the absolute values for the expression of the enzymes.

Considering the content of the diet (Dataset EV10) as well as the calculated bounds for the intracellular reactions in CONV-R and GF mice using RMetD (Dataset EV12), we optimized for the production of chylomicrons and compared the amount of chylomicrons and HDL produced by CONV-R and GF mice (see Materials and Methods). We predicted that lower levels of chylomicrons and HDL are secreted by CONV-R mice compared to GF mice (Fig 7D).

We also revealed the changes in metabolic fluxes in the small intestine of CONV-R and GF in response to gut microbiota, and which of these differences are likely to be associated with transcriptional changes. Through the use *iMouseSmallintestine*, we defined a region of feasible flux distributions using uptake rates for glucose, SCFAs, AAs, and secretion rates for chylomicrons and HDL in CONV-R and GF mice that are predicted by the RMetD. We calculated a set of flux distributions using a random sampling algorithm (Bordel *et al*, 2010) which allowed for identification of transcriptionally regulated reactions. Finally, we calculated the average and standard deviations for each of the fluxes carried by the reactions, and compared the changes in fluxes with the changes in the expression of the genes associated with those reactions. We found that reactions involved in fatty acid biosynthesis and oxidations are transcriptionally down-regulated in CONV-R compared to GF mice (Fig 7E).

## Discussion

The interactions between the gut microbiota, host tissues of the gastrointestinal tract, and diet are known to be highly relevant for

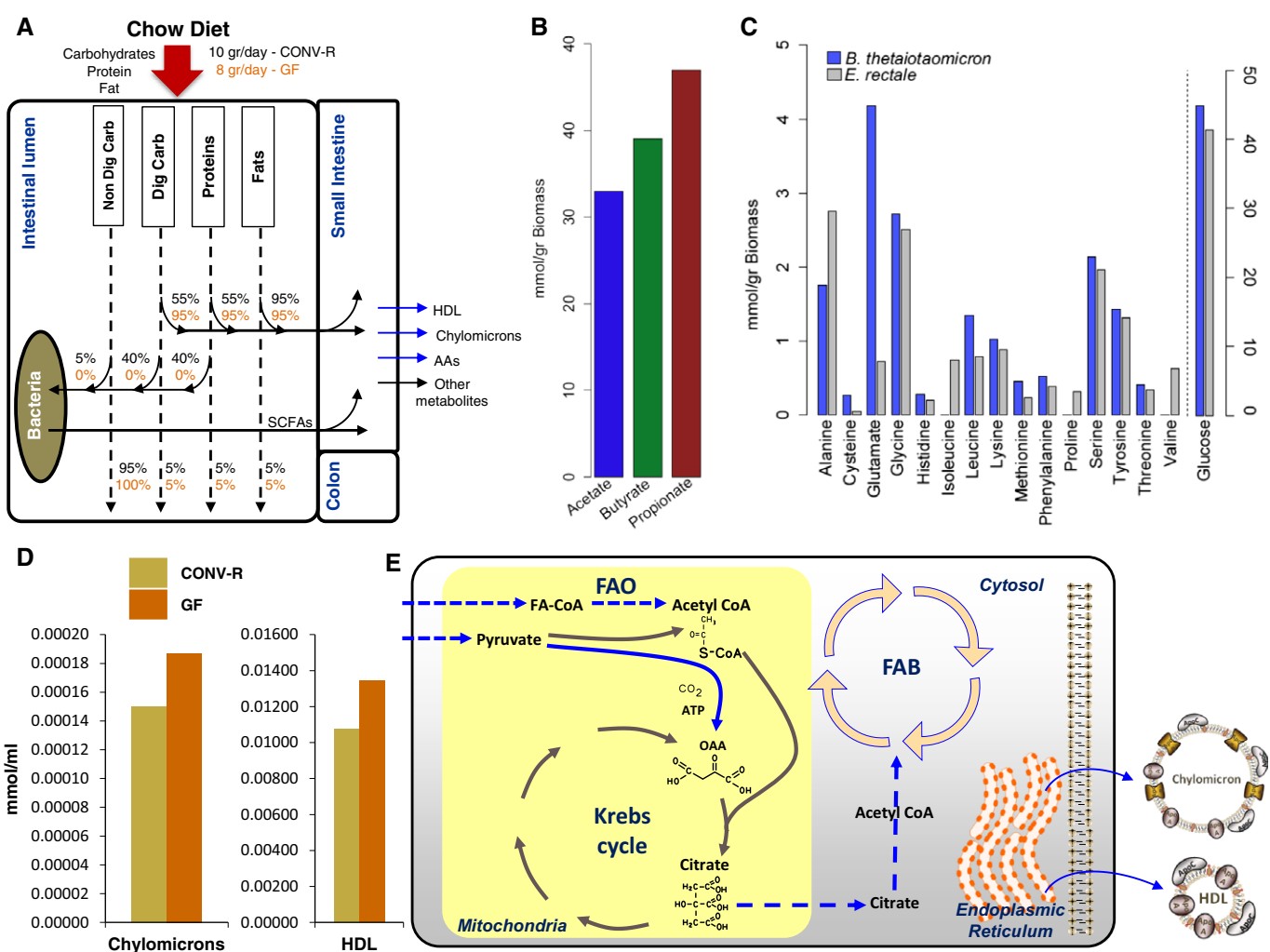

**Figure 7.** *In silico* simulation of the small intestine.

A   The differences in the utilization of proteins, carbohydrates, non-digestible carbohydrates, and fats between the CONV-R and GF mice are presented.

B   The production of the short-chain fatty acids including acetate, butyrate, and propionate produced by the gut microbiota in CONV-R mice is shown.

C   The amount of glucose and AAs as well as their consumption for supporting their biomass is presented.

D   The levels of the chylomicrons and HDL secreted by the small intestine of the CONV-R and GF mice are predicted.

E   Transcriptionally down-regulated (blue arrows) reactions involved in fatty acid biosynthesis and oxidation in small intestine of CONV-R and GF mice are identified through the use of random sampling algorithm.

the health of the host. In order to understand the effect of the gut microbiota on host metabolism, we revealed the metabolic differences between the continuous gastrointestinal tracts as well as liver and WATs of CONV-R and GF mice using tissue-specific GEMs. We observed that the ileum was the tissue that was most affected by the gut microbiota, followed by the duodenum, jejunum, colon, liver, and adipose tissues. Even though, gene expression data for each tissue were analyzed independently, we found that the expression of the Nnt was significantly higher in CONV-R mice compared to GF mice and followed the same directional changes in six of the analyzed tissues.

We validated our GEM-based predictions based on gene expression data by generating metabolomics data, and comparing the level of the metabolites in the PV of the CONV-R and GF mice. Taken together, we found that the levels of the glycine and serine as well

as the N-acetylated form of these two AAs that may be taken up by the liver tissue and used in GSH synthesis were significantly lower in the PV of the CONV-R mice compared to GF mice. Considering the lower expression of the genes involved in glucose uptake as well as the lower level of glycine in the liver, *de novo* synthesis of serine may also be lower in CONV-R mice compared to GF mice. Hence, we observed that the expression of Nnt is increased in the liver of CONV-R mice potentially due to the limited availability of glycine used as a substrate in GSH *de novo* synthesis.

The expression of Nnt is linked to insulin sensitivity, and higher plasma insulin concentrations have been reported in CONV-R mice compared to GF mice (Rabot *et al*, 2010). Naturally occurring deletion of the Nnt in the C57BL/6J mouse strain has been associated with impaired glucose homeostasis control and reduced insulin secretion that is independent of obesity (Freeman *et al*, 2006).

Suppression of Nnt in PC12 rat pheochromocytoma cells led to increased oxidative stress with subsequent impairment of mitochondrial function (Yin *et al*, 2012). Moreover, Ripoll *et al* (2012) overexpressed the NNT in a macrophage cell line and detected decreased levels of reactive oxygen species, indicating that NNT plays a key role in the modulation of the immune response and host defense against pathogens.

The microbiota-induced transcriptional responses in specific fractions of intestinal epithelial cells have also been recently examined by a microarray analysis in CONV-R and GF mice (Sommer *et al*, 2015). It was reported that approximately 10% of the host's transcriptome, mainly genes involved in immune responses, cell proliferation, and metabolism, was regulated by the microbiota.

The differences between the level of free AAs along the GI tract of CONV-R and GF mice have been previously reported (Whitt & Demoss, 1975), and our analysis provided a detailed explanation of how bacteria may regulate the host AA homeostasis. We provided an explanation for the higher expression of the Nnt in CONV-R mice, and observed that the expression of Nnt is increased in CONV-R mice as a response to the decreased level of *de novo* GSH synthesis due to limited availability of glycine. We also found that microbiota in the small intestine may consume glycine as well as other AAs to support its growth and survival which leads to the decreased PV levels of the AAs and regulates the AA and GSH metabolism of the host.

Moreover, we observed that the liver and colon tissues of CONV-R mice also responded to the lower level of glycine by higher expression of Nnt, and this indicated that the gut microbiota regulates AA metabolism not only in the small intestine but also in the liver and the colon. We investigated the global metabolic differences between the liver tissue of the CONV-R and GF mice, and found that the expression of genes involved glucose uptake, and cholesterol and bile acid biosynthesis were significantly lower in CONV-R mice. Previously, it has been shown that the activity and expression of the Cyp7a1, which is the rate-limiting enzyme in bile acid synthesis, is decreased in the liver of CONV-R mice compared to GF mice and that the gut microbiota suppress bile acid synthesis (Sayin *et al*, 2013). Similarly, lower expression of Hmgcr, which is involved in the cholesterol synthesis, has been reported in CONV-R mice compared to GF mice (Sayin *et al*, 2013).

Our analysis also indicated that the gut microbiota regulates the host lipid metabolism. We simulated the metabolic differences between the small intestine of CONV-R and GF mice by integrating relative small intestine gene expression data. We found that lower level of chylomicrons and HDL is produced by the small intestine of the CONV-R. Our RMetD predictions were in agreement with experimental studies in which 40% lower chylomicron levels in CONV-R mice compared to GF mice was detected (Velagapudi *et al*, 2010) and contribution of gut microbiota on the intake of fats was investigated in CONV-R and GF mice (Duca *et al*, 2012).

There are several advantages of RMetD compared to previously developed methods for integrating transcriptomics data into constraint-based models of metabolism (Machado & Herrgard, 2014). Firstly, it only considers enzyme up-/down-regulation rather than their absolute expression. This is based on the assumption that up-/down-regulation of a specific enzyme likely suggests a higher/lower flux in its corresponding reaction. This could be a better solution than inferring fluxes by comparing absolute expression level of different enzymes. In addition, RMetD pushes the flux ranges

instead of the absolute fluxes. Pushing the ranges is more reasonable since in our case (and in most cases), the reference flux distribution of a GEM is unlikely to be exact, but the ranges are much more reliable. Furthermore, multiple objective products could be included in RMetD, and this could be of special interest for those using mammalian tissue GEMs since they usually have more than one obligatory functions and products.

Even though RMetD and other methods (Machado & Herrgard, 2014) for inferring flux rates from gene expression data have been developed, the correlation between the fluxes carried by the reaction and the expression of the gene catalyzing the reaction is known to be limited (Bordel *et al*, 2010). Changes in gene expression levels therefore only serve only as cues for the likelihood that there may be an altered metabolic flux carried by the associated reaction. To validate gene expression data-based predictions, we performed metabolomics analysis in the PV of the both CONV-R and GF mice and these data supported functional changes in the pathways showing altered gene expression.

The functional output and diversity of the gut microbiota are important modulators for the development of various human disorders. Alterations in gut microbiota composition and function have been shown in the pathogenesis of obesity (Ley *et al*, 2006; Turnbaugh *et al*, 2009), T2D (Qin *et al*, 2012; Karlsson *et al*, 2013) and NAFLD (Henao-Mejia *et al*, 2012). Imbalances in the plasma level of glycine as well as other AAs have also been shown in obesity (Newgard *et al*, 2009), T2D (Wang *et al*, 2011a), and NAFLD (Kalhan *et al*, 2011). Strikingly, the plasma levels of glycine are decreased in all subjects with the above-mentioned diseases compared to the healthy subjects. Moreover, we have recently found that the expression of the NNT is significantly increased in both subcutaneous and visceral AT after extensive weight loss in response to bariatric surgery (Mardinoglu *et al*, 2015). In this context, it is of interest to study the microbial AAs in the human GI tract in relation to its potential role in the development of such metabolism-related disorders.

In conclusion, we observed that gut microbiota has a profound systemic effect on AA and GSH as well as lipid metabolism, and it is one of the major regulators of metabolism in mammals. We demonstrate that a detailed understanding of the metabolic differences between CONV-R and GF mice obtained through GEM modeling may allow for revealing the key roles of the gut microbiota. Our findings may be used for investigating the contribution of the gut microbiome in the progression of metabolism-related disorders as well as for elucidating the unknown etiology of such disorders.

# Materials and Methods

### Mice

Male C57Bl6/J mice aged 12–14 weeks were used in these experiments. CONV-R mice were housed in individually ventilated cages, and GF mice were housed in flexible film isolators, with maximum 5 mice per cage, and fed a normal chow *ad libitum*. GF status was verified regularly by anaerobic culturing in addition to PCR for bacterial 16S rDNA. Light cycle was 12/12 h dark/light and lights on at 06:00. Mice were euthanized by cervical dislocation, and tissues were collected in liquid nitrogen immediately. All procedures were

approved by the Gothenburg University Ethical Committee (Permit No. 339/2012).

RNA from the liver as well as epididymal and subcutaneous WATs obtained from CONV-R and GF mice were isolated using the RNeasy Lipid Tissue Mini Kit (Qiagen, Hilden, Germany). RNA concentration and quality were evaluated by spectrophotometric analysis (ND-1000; NanoDrop Technologies, Wilmington, DE, USA) and capillary electrophoresis on a 2100 Bioanalyzer (Agilent Technologies, Santa Clara, CA, USA). Tissue samples with no quality issues were used in the generation of the gene expression data.

## Transcriptomics data

Global gene expression profiling of liver as well as epididymal and subcutaneous WATs was measured using MoGene 1.0 ST chips (Affymetrix). CEL files for duodenum, jejunum, ileum, and colon tissue of CONV-R and GF mice were retrieved from Gene Expression Omnibus (GEO) public repository under the accession number GSE17438.

Raw data for each tissue were preprocessed independent of the other tissues and normalized with robust multi-array average (RMA) using Piano R package (Väremo *et al*, 2013). Probes which are not mapped to any gene from the analysis were removed, and the differential expression analysis for 23,736 probe sets between CONV-R and GF mice was carried out by calculating two-way analysis of variance (ANOVA) *P*-values. *P*-values were adjusted for multiple testing using the R-function p.adjust with the method set to false discovery rate (FDR), and *Q*-values were calculated. *Q*-values were used for network-dependent analysis.

During the simulation of the metabolic differences between the small intestine of CONV-R and GF mice, gene expression data for duodenum, jejunum, and ileum were integrated. However, acyl-CoA dehydrogenase, long chain (Acadl), cell division cycle 14A (Cdc14a), cystic fibrosis transmembrane conductance regulator (Cftr), cathepsin B (Ctsb), fructose-1,6-bisphosphatase 1 (Fbp1), and 6-phosphofructo-2-kinase/fructose-2,6-biphosphatase 3 (Pfkfb3), which were significantly changed (*Q*-value < 0.05) in different directions between the segments of the small intestine, were not included into our analysis.

Raw data for liver as well as epididymal and subcutaneous WATs will be deposited in GEO public repository under the accession number GSE31115. To make the gene expression data easily available to the scientific community, the data are also included into the searchable database (http://microbiota.wall.gu.se).

## RT–PCR

Liver tissue samples obtained from CONV-R and GF mice were dissected out and immediately frozen in liquid nitrogen. Total RNA was extracted from frozen liver tissue samples using RNeasy Mini Kit (Qiagen, Hilden, Germany). About 0.5 μg of RNA was used for cDNA synthesis, using High Capacity cDNA reverse transcription kit (Applied Biosciences). Primers were obtained from Sigma, and specificity was verified by Blast and the appearance of a single product band of predicted size. qRT–PCR was performed in a CFX96 Real-time System (Bio-Rad). Expression levels of Nnt and Entpd4 were calculated relative to the mRNA expression of L32 and calculated with the $\Delta\Delta\text{-}C_t$ method.

## Reconstruction of mouse tissue-specific GEMs

SILAC-based proteome reflected the proteins expressed in the specified tissues, and in total, 7,349 proteins were analyzed in 28 different major C57BL/6 mouse tissues. We reconstructed tissue-specific GEMs for each tissue using the proteomics data (Geiger *et al*, 2013) and recently developed tINIT algorithm (Agren *et al*, 2014). During the reconstruction of the 28 draft GEMs, we used 56 common tasks that are known to occur in all human cells (Agren *et al*, 2014). These metabolic functions include the provision of energy and redox, utilization, and internal conversion substrates as well as the biosynthesis of certain metabolites. We compared these automated tissue-specific GEMs and found that GEM for liver contained the largest number of tissue-specific reactions, metabolites, and genes, which represents its unique role in overall host metabolism. Moreover, we reconstructed functional GEMs for liver and adipose tissue using the previously developed human models (Mardinoglu *et al*, 2013a, 2014a,b) and functional GEMs for small intestine and colon tissues. It should be noted that we reconstructed a functional generic GEM for small intestine rather than specific GEMs for duodenum, jejunum, and ileum.

## AA and N-acetylated AA level in CONV-R and GF mice

In order to validate our model-based predictions, we raised six CONV-R and six GF C57Bl6/J mice and fed with autoclaved chow diet. We collected blood samples from hepatic portal vein after 12 weeks and detected the level of the AAs and acteyl-AAs. Samples were prepared using the automated MicroLab STAR® system from Hamilton Company. A recovery standard was added prior to the first step in the extraction process for QC purposes. To remove protein, dissociate small molecules bound to protein or trapped in the precipitated protein matrix, and to recover chemically diverse metabolites, proteins were precipitated with methanol under vigorous shaking for 2 min (Glen Mills GenoGrinder 2000) followed by centrifugation.

The liquid chromatography-tandem mass spectrometry (LC-MS/MS) portion of the platform was based on a Waters ACQUITY ultra-performance liquid chromatography (UPLC) and a Thermo-Finnigan LTQ mass spectrometer operated at nominal mass resolution, which consisted of an electrospray ionization (ESI) source and linear ion-trap (LIT) mass analyzer. The sample extract was dried and then reconstituted in acidic or basic LC-compatible solvents, each of which contained 12 or more injection standards at fixed concentrations. One aliquot was analyzed using acidic positive ion-optimized conditions and the other using basic negative ion-optimized conditions in two independent injections using separate dedicated columns (Waters UPLC BEH C18-2.1 × 100 mm, 1.7 μm). Extracts reconstituted in acidic conditions were gradient eluted using water and methanol containing 0.1% formic acid, while the basic extracts, which also used water/methanol, contained 6.5 mM ammonium bicarbonate. The MS analysis alternated between MS and data-dependent MS/MS scans using dynamic exclusion, and the scan range was from 80 to 1,000 m/z.

Following log transformation and imputation of missing values, if any, with the minimum observed value for each compound, Welch's two-sample *t*-test was used to identify AAs and N-acetylated AAs that differed significantly between CONV-R and GF mice, determined independently for each dataset. A summary of

the numbers of biochemicals that achieved statistical significance ($P \leq 0.05$) is shown in Figs 3D and 4A and C. An estimate of the false discovery rate ($Q$-value) is calculated to take into account the multiple comparisons that normally occur in metabolomics-based studies.

### Integration of gene expression data into GEM for small intestine using RMetD

In order to integrate gene expression data into GEMs, we developed relative metabolic differences (RMetD), which allow for the application of relative gene expressions between CONV-R and GF mice rather than the absolute values, and simulated the metabolic differences using the content of the diet (Dataset EV10). First, we set the lower bounds of production of HDL and chylomicrons to 20% (arbitrary value) of their maximum production in GF small intestine model. We performed flux variability analysis for all reactions associated with the significantly ($Q$-value < 0.05) differentially expressed genes in the model, and found the upper and lower bound of these reactions. A reaction could be associated with more than one gene, and these genes may have different expression trends (e.g. one gene is up-regulated, whereas the other genes are down-regulated). In such cases, we assumed that genes associated with these reactions are not significantly changed ($Q$-value < 0.05).

Next, for reactions associated with the up-regulated genes in CONV-R mice, we set both the upper and lower bounds of the reactions in CONV-R model 20% (arbitrary value) more than the bound of the reactions in GF model whereas 20% less for the reactions associated with the down-regulated genes. By this way, reactions with up-/down-regulated gene expression were able to carry more/ less fluxes. By adding these new constraints for each reaction in CONV-R and GF mice together with the use of the content of the diet, we predicted the fluxes of both models by maximizing the production of chylomicrons and minimizing the sum of fluxes. For all arbitrary values, we performed sensitivity analysis by using different values and obtained similar results.

All the simulations were carried out using RAVEN toolbox (Agren *et al*, 2013). RMetD source code was implemented in MATLAB and RAVEN toolbox, and it is publically available at https://sourceforge.net/projects/relative-metabolic-differences/files/ MiceStudy/.

### Data availability

MMR generic mouse GEM, and functional GEMs for liver, adipose, colon, and small intestine as well as the other mouse tissue GEMs are publically available in the Systems Biology Mark-up Language (SBML) format at Human Metabolic Atlas (http://www.metabolicatlas.org) and at BioModels database with accession numbers MODEL1509220000–MODEL1509220032. They are also provided as Computer Code EV1.

Gene expression data from liver as well as epididymal and subcutaneous WATs of CONV-R and GF mice are publically available in Gene Expression Omnibus (GEO) database with the accession number GSE31115.

**Expanded View** for this article is available online: http://msb.embopress.org

## Acknowledgements

This project was supported by the Bill & Melinda Gates Foundation, Knut and Alice Wallenberg Foundation, Torsten Söderbergs Stiftelse, and European Commission FP7 project METACARDIS with the grant agreement HEALTH-F4-2012-305312, Novo Nordisk A/S, and Novo Nordisk Foundation. We would like to thank Prof. Jan Borén, University of Gothenburg, for providing constructive comments and Doruk Demircioglu for scientific illustrations.

## Author contributions

AM created MMR, generated tissue-specific models, and performed the analysis of the gene expression data together with JN. SS performed the *in silico* simulations between the bacteria. MB, EL, and FB generated gene expression data for liver and adipose tissues, performed the PCR analysis, and measured the level of amino acids and N-acetylated amino acids. PG compared the tissue-specific models. CZ developed RMetD and performed the simulations of the small intestine. AM and JN conceived the project.

## Conflict of interest

The authors declare that they have no conflict of interest.

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
