## [Review Process File · Molecular Systems Biology]

The gut microbiota modulates host amino acid and glutathione metabolism in mice

Adil Mardinoglu, Saeed Shoaie, Mattias Bergentall, Pouyan Ghaffari, Cheng Zhang, Erik Larsson, Fredrik Bäckhed and Jens Nielsen

Corresponding author: Adil Mardinoglu, Chalmers University of Technology

Review timeline:

Submission date:	05 August 2015
Editorial Decision:	11 September 2015
Revision received:	23 September 2015
Accepted:	25 September 2015

Editor: Maria Polychronidou

Transaction Report:

1st Editorial Decision

11 September 2015

Thank you again for submitting your work to Molecular Systems Biology. We have now heard back from the two referees who agreed to evaluate your manuscript. As you will see from the reports below, the referees think that the presented findings seem interesting. However, they raise a series of concerns, which should be carefully addressed in a revision of the manuscript. The referees' recommendations are rather clear and therefore there is no need to repeat the points listed below.

If you feel you can satisfactorily deal with these points and those listed by the referees, you may wish to submit a revised version of your manuscript. Please attach a covering letter giving details of the way in which you have handled each of the points raised by the referees. A revised manuscript will be once again subject to review and you probably understand that we can give you no guarantee at this stage that the eventual outcome will be favorable.

REFeree REPORTS

Reviewer #1:

Summary

The authors use germfree mice to explore the contribution of the microbiota to mouse metabolism. By integrating the gene expression information from previously profiled intestinal tissue and newly collected liver and adipose tissue, the authors were able to find an amino acid metabolism gene (Nnt) that was significantly upregulated in all relevant tissues of conventionally raised as compared to germfree mice. The authors take a systems level approach to explore the biological relevance of this gene expression change by constructing mouse metabolic reaction genome scale metabolic models (MMR) and by modeling the microbial utilization of dietary amino acids. These models support the concept that increased Nnt levels are a compensatory mechanism to sustain glutathione

(GSH) levels in situations of decreased de novo synthesis of GSH due to microbial depletion of dietary glycine in the conventionally raised state. Ultimately the authors were able to validate some of their predictions through metabolomics studies that showed lower levels of glycine and serine in the hepatic portal vein of conventionally raised mice. This paper also presents a significant contribution to the understanding of general host metabolism through the development of Relative metabolism differences (RmetD) which uses relative instead of absolute changes in gene expression to model differences in metabolic pathways between two states. Through RmetD the authors were able to capitalize on the strength of their multiple gene expression data sets to determine that there CONVR mice have lower expression of fatty acid biosynthesis genes in CONVR mice.

General Remarks

- This paper presents a conceptual advance on the microbial contribution to glutathione metabolism (the host upregulates Nnt to compensate for a diminished ability to de novo synthesize GSH in the conventionally raised state)
- This paper presents 4 new biological datasets (liver, epididymal fat, and subcutaneous fat gene expression as well as hepatic portal vein metabolomics) that are publically available
- This paper contributes some valuable framework (MMRs and tissue specific MMRs) available for public use at the human metabolic atlas site
- Present ways to extract meaningful metabolism specific information from the integration of multiple gene expression datasets in the creation of RMetD

Key audience for this work:

- Microbiome community
- Investigators interested in extracting biologically relevant information from large scale gene expression data sets from multiple tissues

Major Points

1. are the lack of expression differences in adipose tissue due to the difficult nature of RNA isolation/collection procedures in this tissue or to biological similarity of gene expression between GF and CONVR mice in adipose tissue

Is there an issue with collecting adipose tissue that would affect the isolation of RNA and subsequent detection of differential gene expression? Was the quality of the adipose tissue RNA comparable to the other collected tissues?

2. are the human GEMs and mouse GEMs comparable?

Is it fair to compare mouse and human GEMs and claim difference in heterogeneity when the construction of the mouse GEM depended on the human GEM? -p.6

Is the proteomics data for humans as comprehensive as the proteomics data for mice and does this compromise the ability to compare MMR and the human GEMs?

If the MMR is based on mouse orthologs of human genes - what happens to mouse genes that don't have a human ortholog? What percentage of mouse genes are left out of the model due to unmatched mouse genes?

3. lack of sense of effect size when gene expression changes are solely represented as in a binary metric as low or high.

At least for the 2 genes with significant expression changes (Nnt and Entpd4) throughout the majority of tissue types the magnitude of gene expression changes should be displayed in heatmap or bar chart format

4. What is the purpose of briefly mentioning 10 of the 29 metabolism genes differentially expressed in GF vs. CONVR mouse livers? Why those 10 genes?

5. wording of statement on p.8 strong statement that central metabolism is higher in CONVR mice. Great analysis of metabolism specific gene expression but strong (maybe too strong) statements of overall metabolism based on gene expression. I would agree that the overall expression of central metabolism genes is downregulated in the colons of CONVR mice. Gene expression only accounts for a limited set of knowledge about the system. Metabolic outcomes are driven by protein expression and enzymatic activity which have not been directly profiled in all tissue analyzed for gene expression in this paper - Reference reviewing how mRNA expression and protein expression

are correlated ~40% of the time depending on the system.
<http://www.nature.com/nrg/journal/v13/n4/full/nrg3185.html#B1>

Would be a valuable addition for the authors to directly show experimentally that de novo GSH synthesis is down in CONV-R mice but probably out of the scope of this paper

Minor Points

What is the meaning of this statement? - "Our results indicated that the gut microbiota associated with the altered AA metabolism of the host"

- p.3 "previously published gene expression profile(s)" - add (s)
- p.3 "conventionally raised (CR)" should read "conventionally raised (CONV-R)" to follow the conventions of the rest of the paper
- p.3 - is 2012 really recently?
- p.5 "In average" changed to "On average"
- p.6 The "GR" abbreviation should read "GF"
- p.6 "in ileum" should read "in the ileum"
- p.10 "de novo synthesis of (the) serine" -remove "the"

Figure 1A is not a useful infographic - something detailing the anatomy of the organs profiled and including the location of the hepatic portal vein may be of better use to the reader

Figure 3D, 4A, 5B - y-axis label needed

Is it ileum tissue or ileal tissue?

Reviewer #2:

The manuscript by Mardinoglu et al. presents an interesting genome scale analysis of the ability of the gut microbiome to modulate the metabolism of amino acids, glutathione, and lipoprotein, by integrating tissue specific transcriptome and metabolome information with host and microbe metabolic models. Along with previously collected transcriptomes from the intestine, the authors analyzed the transcriptomes from Germ Free (GF) and Conventionally Raised (Conv-R) mouse liver and fatty tissue. Notably, GF and Conv-R transcriptomes separate by PCA of most tissues. Subsequently, differentially expressed metabolic genes were identified, with Nnt and Entpd4 being significantly differentially expressed across intestinal and liver tissue. The authors generate a mouse metabolic reconstruction by projecting mouse homologs onto the human reconstruction HMR2, and used SILAC data with the tINIT algorithm to generate tissue specific metabolic models. Using these reconstructions to determine genes interacting with Nnt, they identify significant alterations in the metabolism of glutathione and amino acid metabolic pathways as in GF mice. The authors next quantified the availability of amino acids in the hepatic portal vein (HPV), and verified that the abundances of many amino acids (as well as their N-acetylated counterparts) are reduced in Conv-R mice. Finally, metabolic modeling of an idealized gut microbiome revealed specific modes of consumption of and exchange of amino acids by the microbiome, and metabolic modeling of differential activity of the mouse gut predicted alteration of lipoprotein secretion in GF mice. The work presents an advanced integrative analysis of multi-omic data and modeling in a model experimental system. It describes specific a specific role the microbiota plays in maintaining host health and illustrates a potential mechanism for its behavior. While the research is creative and presents impressive work, the manuscript can be improved in a number of ways to better highlight the findings described:

General comments:

1. The specific methodologies employed for particular analyses, or the motivation for such methods, is often unclear. See below for specific examples.

Specific comments:

1. The distinction between specific models is unclear. Specifically, it appears that the MMR is distinguished as simply a reconstruction of general mouse metabolism, but what is the difference

between the GEMs described by figure 2 (and for which heterogeneity degree is calculated) and the four functional GEMs (e.g., iMiceSmallintestine)?

2. In line with the above, a more detailed description of the methods used to generate these models is warranted. For example, a brief description of the tINIT algorithm should be given: What is the basic operation and objective of the algorithm itself, not just the parameters utilized.

3. Please give a basic description of what is meant by 'heterogeneity degree' of GEMs, as well as the calculation involved. Additionally, please show additional support of your claim that "mouse metabolism can carry slightly more complex metabolism using fewer enzymes". This may be an artifact of the methodology, given that the reconstruction process begins with a pruning of the human enzyme complement, with a gap filling step on metabolites (and the edges connecting them in the metabolic network). Additionally, this claim, while interesting, may be more suited to the discussion section of the manuscript.

4. It is unclear what advantage the network provides in identifying differentially expressed genes (or what information it provides). The results section begins with a large scale analysis of differential expression which identifies specific genes (Nnt & Entpd4), but only after the GEMs are introduced is the claim made that (e.g.) Gsr is differentially expressed. Is this because Gsr is differential in some but not all of the analyzed tissues, or perhaps is the network used to reduce the search space of genes of interest, with a commensurate reduction in the need to perform stringent multiple hypothesis correction? Please clarify.

5. It is unclear why the authors chose to focus on chylomicrons. Were additional molecules investigated but not revealed to be significant, or (perhaps more likely), is there a specific biological question being raised: if so, please make this question explicit.

6. Please clarify your method for performing multiple hypotheses correction. The Benjamini-Hochberg method provides a False Discovery Rate, while Storey's method provides q-values. Which method (or methods) is used (or how they are combined) is unclear.

Additional minor comments:

1. The manuscript can be improved by a thorough proof-reading. Errors in grammar and spelling appear throughout.

2. Please do not break clauses with parentheses. For example, consider "significantly higher (Q-value<0.05)" in place of "significantly (Q-value<0.05) higher".

1st Revision - authors' response

23 September 2015

Response to Reviewers

Reviewer #1:

Summary

The authors use germfree mice to explore the contribution of the microbiota to mouse metabolism. By integrating the gene expression information from previously profiled intestinal tissue and newly collected liver and adipose tissue, the authors were able to find an amino acid metabolism gene (Nnt) that was significantly upregulated in all relevant tissues of conventionally raised as compared to germfree mice. The authors take a systems level approach to explore the biological relevance of this gene expression change by constructing mouse metabolic reaction genome scale metabolic models (MMR) and by modeling the microbial utilization of dietary amino acids. These models support the concept that increased Nnt levels are a compensatory mechanism to sustain glutathione (GSH) levels in situations of decreased de novo synthesis of GSH due to microbial depletion of dietary glycine in the conventionally raised state. Ultimately the authors were able to validate some of their predictions through metabolomics studies that showed lower levels of glycine and serine in the hepatic portal vein of conventionally raised mice. This paper also presents a significant contribution to the understanding of general host metabolism through the development of Relative metabolism differences (RmetD) which uses relative instead of absolute changes in gene expression to model differences in metabolic pathways between two states. Through RmetD the authors were

able to capitalize on the strength of their multiple gene expression data sets to determine that there CONVR mice have lower expression of fatty acid biosynthesis genes in CONVR mice.

General Remarks

- This paper presents a conceptual advance on the microbial contribution to glutathione metabolism (the host upregulates Nnt to compensate for a diminished ability to de novo synthesize GSH in the conventionally raised state)
- This paper presents 4 new biological datasets (liver, epididymal WAT, and subcutaneous WAT gene expression as well as hepatic portal vein metabolomics) that are publically available
- This paper contributes some valuable framework (MMRs and tissue specific MMRs) available for public use at the human metabolic atlas site
- Present ways to extract meaningful metabolism specific information from the integration of multiple gene expression datasets in the creation of RMetD

Key audience for this work:

- Microbiome community
- Investigators interested in extracting biologically relevant information from large scale gene expression data sets from multiple tissues

We wish to thank Reviewer #1 for detailed reading of our manuscript and providing constructive comments. We would like to address the specific comments as follows:

Major Points

1. are the lack of expression differences in adipose tissue due to the difficult nature of RNA isolation/collection procedures in this tissue or to biological similarity of gene expression between GF and CONVR mice in adipose tissue

Is there an issue with collecting adipose tissue that would affect the isolation of RNA and subsequent detection of differential gene expression? Was the quality of the adipose tissue RNA comparable to the other collected tissues?

As reviewer mentioned, it is generally difficult to isolate RNA from adipose tissue compared to other tissues such as liver and gut tissues. To ensure high quality, we used RNeasy Lipid Tissue Mini Kit (Qiagen, Hilden, Germany) for isolating RNA from liver, adipose and gut tissues. Next, we evaluated RNA concentration and quality by spectrophotometric analysis (ND-1000; NanoDrop Technologies, Wilmington, DE, USA) and capillary electrophoresis on a 2100 Bioanalyzer (Agilent Technologies, Santa Clara, CA, USA). RNA isolated from all tissues was comparable to each other and all passed the quality control before the generation of the gene expression data. Quality control also revealed no quality issues with the adipose tissue samples – if anything quality was better in these samples (all included samples were of high quality).

After multiple testing of the gene expression data, we found seven significantly differentially expressed probesets in subcutaneous WAT. Even though, we analyzed microarray data for each tissue independently, we found that that Nnt and Entpd4 were significantly different in subcutaneous WAT as well as in liver and gut tissues. On the other hand, we could not detect any differences between the Epididymal WAT after multiple testing.

Considering that GF and CONV-R mice were both lean, we think our gene expression data represent the real biological differences. We also included the following statement in the M&M.

"Male C57Bl6/J mice aged 12-14 weeks were used in these experiments. CONV-R mice were housed in individually ventilated cages and GF mice were housed in flexible film isolators, with maximum 5 mice per cage, and fed a normal chow ad libitum. GF status was verified regularly by anaerobic culturing in addition to PCR for bacterial 16S rDNA. Light cycle was 12/12 h dark/light and lights on at 06:00. Mice were dissected by cervical dislocation and tissues were collected in liquid nitrogen immediately. All procedures were approved by the Gothenburg University ethical committee (Permit No. 339/2012).

RNA from the liver as well as epididymal and subcutaneous WATs obtained from CONV-R and GF mice was isolated using the RNeasy Lipid Tissue Mini Kit (Qiagen, Hilden, Germany). RNA concentration and quality was evaluated by spectrophotometric analysis (ND-1000; NanoDrop

Technologies, Wilmington, DE, USA) and capillary electrophoresis on a 2100 Bioanalyzer (Agilent Technologies, Santa Clara, CA, USA). Tissue samples with no quality issues were used in the generation of the gene expression data."

2. are the human GEMs and mouse GEMs comparable?

We analyzed the heterogeneity of the mouse tissue-specific GEMs (Figure 2D) based on incorporated reactions, genes and metabolites since all GEMs were reconstructed using the same list of metabolic functions and SILAC based proteomics data with the same coverage for each tissue. Next, we analyzed the heterogeneity of the recently generated human cell types which are reconstructed using the exact same list of metabolic functions and antibody based proteomics data. Finally, we presented the heterogeneity degree of mouse tissues and human cell types separately. In order to avoid confusion, we rewrote the related section as below.

"We analyzed the heterogeneity of the mouse tissue-specific GEMs (Figure 2D) based on incorporated reactions, genes and metabolites by calculating the heterogeneity degree of the models. The heterogeneity degree allowed us to capture the divergence between metabolic networks based on their constituent parameters including reactions, metabolites and genes and it was calculated using the average and maximum Hamming distance of the model (Ghaffari et al, 2015). Moreover, we analyzed the heterogeneity of recently reconstructed human cell-specific GEMs (Agren et al, 2014) that have been reconstructed based on antibody based proteomics data in the Human Protein Atlas (www.proteinatlas.org) (Uhlen et al, 2015; Uhlen et al, 2010). On average, mouse tissue-specific GEMs showed an average heterogeneity degree of 0.77 for reactions, 0.72 for metabolites and 0.78 for genes whereas human cell-specific GEMs had an average heterogeneity degree of 0.8 for reactions, 0.7 for metabolites and 0.84 for genes (Figure 2D). Compared with the human cell-specific GEMs, the mouse tissue-specific GEMs had a slightly higher metabolic uniformity and lower heterogeneity based on the incorporated genes and reactions, but they had a slightly higher heterogeneity based on the incorporated metabolites into the models."

Is it fair to compare mouse and human GEMs and claim difference in heterogeneity when the construction of the mouse GEM depended on the human GEM? -p.6

This is a good point, but as the individual models are generated based on actual data from mouse and human, we believe that it is fair to make this comparison. The generic model is practical identical for the two organisms, but as different proteins may be expressed in the same tissues of the two organisms there may be some differences in the tissue specific GEMs and this is what we focus our analysis on.

Is the proteomics data for humans as comprehensive as the proteomics data for mice and does this compromise the ability to compare MMR and the human GEMs?

Human cell specific-specific GEMs have been reconstructed based on the proteomics data in Human Protein Atlas (version 12) and it covers approximately 3,200 of the 3,765 protein coding genes in HMR2. On the other hand SILAC based proteomics data for mouse tissues cover 2,030 of the 3,580 protein-coding genes in MMR. As we mentioned above, rather than comparing mouse tissue-specific and human cell-specific GEMs, we intended to present the differences within the mouse and human GEMs independently.

If the MMR is based on mouse orthologs of human genes - what happens to mouse genes that don't have a human ortholog? What percentage of mouse genes are left out of the model due to unmatched mouse genes?

We constructed MMR by using the mouse orthologs of human genes in HMR2 and the resulting generic model includes 8,140 metabolism related reactions, 3,579 associated metabolic genes. We removed genes and associated reactions from the MMR when we could not find the mouse orthologs of human genes. As we have 3,765 genes in HMR2, we only removed about 200 genes from human model. It is of course likely that mouse may have some metabolic genes that we are not capturing using our approach, but we expect these to be of minor importance.

3. lack of sense of effect size when gene expression changes are solely represented as in a binary metric as low or high.

At least for the 2 genes with significant expression changes (Nnt and Entpd4) throughout the majority of tissue types the magnitude of gene expression changes should be displayed in heatmap or bar chart format

We have presented the differentially expressed probesets and corresponding p-values, q-values, log₂ fold changes as well as the genes for each tissue in the Dataset EV1. We agree with the reviewer that magnitude of gene expression changes as well as the corresponding p-values should be available for others.

4. What is the purpose of briefly mentioning 10 of the 29 metabolism genes differentially expressed in GF vs. CONVR mouse livers? Why those 10 genes?

As we mentioned, we presented all of the significantly differentially expressed liver tissue genes in Dataset EV1. Moreover, we presented 13 of these genes in Figure 5 since these genes are connected in a smaller network of the liver tissue GEM.

5. wording of statement on p.8 strong statement that central metabolism is higher in CONVR mice.

Based on the suggestion of the reviewer, we revised this statement.

Great analysis of metabolism specific gene expression but strong (maybe too strong) statements of overall metabolism based on gene expression. I would agree that the overall expression of central metabolism genes is downregulated in the colons of CONVR mice. Gene expression only accounts for a limited set of knowledge about the system. Metabolic outcomes are driven by protein expression and enzymatic activity which have not been directly profiled in all tissue analyzed for gene expression in this paper - Reference reviewing how mRNA expression and protein expression are correlated ~40% of the time depending on the system.

<http://www.nature.com/nrg/journal/v13/n4/full/nrg3185.html#B1>

We were able to validate our gene expression data based predictions through the use of metabolomics data and this generated more confidence in our study. However, we also agree with the comment of the reviewer that gene expression data cannot be used to predict the fluxes carried by the associated reactions. Based on the suggestion of the reviewer, we now included the following sentences to the discussion part of the paper.

"Even though RMetD and other methods (Machado & Herrgard, 2014) for inferring flux rates from gene expression data have been developed, the correlation between the fluxes carried by the reaction and the expression of the gene catalyzing the reaction is known to be limited (Bordel et al, 2010). Changes in gene expression levels therefore only serve only as cues for the likelihood that there may be an altered metabolic flux carried by the associated reaction. To validate gene expression data based predictions we performed metabolomics analysis in the PV of the both CONV-R and GF mice and these data supported functional changes in the pathways having altered gene expression."

Would be a valuable addition for the authors to directly show experimentally that de novo GSH synthesis is down in CONV-R mice but probably out of the scope of this paper.

We agree with the reviewer that it would be valuable to show that de novo GSH synthesis is downregulated in CONV-R mice but it is out of the scope of our study as reviewer acknowledged.

Minor Points

We thank the reviewer for the minor issues. We have now fixed all in the revised version of the manuscript.

What is the meaning of this statement? - "Our results indicated that the gut microbiota associated with the altered AA metabolism of the host"

We rewrote as "Our results indicated that the gut microbiota alter AA metabolism of the host."

p.3 "previously published gene expression profile(s)" - add (s)

p.3 "conventionally raised (CR)" should read "conventionally raised (CONV-R)" to follow the conventions of the rest of the paper

p.3 - is 2012 really recently?

p.5 "In average" changed to "On average"

p.6 The "GR" abbreviation should read "GF"

p.6 "in ileum" should read "in the ileum"

p.10 "de novo synthesis of (the) serine" -remove "the"

Figure 1A is not a useful infographic - something detailing the anatomy of the organs profiled and including the location of the hepatic portal vein may be of better use to the reader

Based on the suggestion of the reviewer, we updated Figure 1A.

Figure 3D, 4A, 5B - y-axis label needed

Based on the suggestion of the reviewer, we added y-axis labels to the figures.

Is it ileum tissue or ileal tissue?

It is ileum tissue.

Reviewer #2:

The manuscript by Mardinoglu et al. presents an interesting genome scale analysis of the ability of the gut microbiome to modulate the metabolism of amino acids, glutathione, and lipoprotein, by integrating tissue specific transcriptome and metabolome information with host and microbe metabolic models. Along with previously collected transcriptomes from the intestine, the authors analyzed the transcriptomes from Germ Free (GF) and Conventionally Raised (Conv-R) mouse liver and fatty tissue. Notably, GF and Conv-R transcriptomes separate by PCA of most tissues. Subsequently, differentially expressed metabolic genes were identified, with Nnt and Entpd4 being significantly differentially expressed across intestinal and liver tissue. The authors generate a mouse metabolic reconstruction by projecting mouse homologs onto the human reconstruction HMR2, and used SILAC data with the tINIT algorithm to generate tissue specific metabolic models. Using these reconstructions to determine genes interacting with Nnt, they identify significant alterations in the metabolism of glutathione and amino acid metabolic pathways as in GF mice. The authors next quantified the availability of amino acids in the hepatic portal vein (PV), and verified that the abundances of many amino acids (as well as their N-acetylated counterparts) are reduced in Conv-R mice. Finally, metabolic modeling of an idealized gut microbiota revealed specific modes of consumption of and exchange of amino acids by the microbiome, and metabolic modeling of differential activity of the mouse gut predicted alteration of lipoprotein secretion in GF mice.

The work presents an advanced integrative analysis of multi-omic data and modelling in a model experimental system. It describes specific a specific role the microbiota plays in maintaining host health and illustrates a potential mechanism for its behaviour. While the research is creative and presents impressive work, the manuscript can be improved in a number of ways to better highlight the findings described:

General comments:

1. The specific methodologies employed for particular analyses, or the motivation for such methods, is often unclear. See below for specific examples.

Specific comments:

1. The distinction between specific models is unclear. Specifically, it appears that the MMR is distinguished as simply a reconstruction of general mouse metabolism, but what is the difference between the GEMs described by figure 2 (and for which heterogeneity degree is calculated) and the four functional GEMs (e.g., iMiceSmallintestine)?

We first constructed MMR, a generic mouse model using the mouse orthologs of human genes in HMR2 and generated mouse tissue-specific GEMs based on the SILAC based proteomics data. In Figure 2, we presented the differences within the mouse tissue-specific GEMs which were reconstructed using the tINIT algorithm and proteomics data. However, we did not include the manually curated GEMs in to the analysis since they were reconstructed using different methodology. We think it is not fair to compare the manually reconstructed GEMs with automatically reconstructed GEMs. We have now clarified this in the text.

2. In line with the above, a more detailed description of the methods used to generate these models is warranted. For example, a brief description of the tINIT algorithm should be given: What is the basic operation and objective of the algorithm itself, not just the parameters utilized.

Based on the suggestion of the reviewer, we now included the statement below.

"The tINIT algorithm allows for the reconstruction of functional GEMs based on the global proteomics data as well as user defined metabolic tasks, which the resulting model should be able to perform."

3. Please give a basic description of what is meant by 'heterogeneity degree' of GEMs, as well as the calculation involved. Additionally, please show additional support of your claim that "mouse metabolism can carry slightly more complex metabolism using fewer enzymes". This may be an artifact of the methodology, given that the reconstruction process begins with a pruning of the human enzyme complement, with a gap filling step on metabolites (and the edges connecting them in the metabolic network). Additionally, this claim, while interesting, may be more suited to the discussion section of the manuscript.

We clarified the entire section about the heterogeneity degree and rewrote it in the revised version manuscript. Please see our response to Reviewer 1.

We intended to analyze the heterogeneity of the mice tissue-specific and human cell-specific GEMs within each other rather than comparing between mouse and human GEMs. It is therefore, we removed our statement "This may indicate that mouse metabolism can carry slightly more complex metabolism using fewer enzymes in the different tissues." from the revised version of the manuscript. With such relatively small differences in the heterogeneity degrees, it may be inappropriate to make this claim.

4. It is unclear what advantage the network provides in identifying differentially expressed genes (or what information it provides). The results section begins with a large scale analysis of differential expression which identifies specific genes (Nnt & Entpd4), but only after the GEMs are introduced is the claim made that (e.g.) Gsr is differentially expressed. Is this because Gsr is differential in some but not all of the analyzed tissues, or perhaps is the network used to reduce the search space of genes of interest, with a commensurate reduction in the need to perform stringent multiple hypothesis correction? Please clarify.

Here, we used the network topology provided by the tissue-specific GEMs in the analysis of the gene expression data. As reviewer acknowledged, Nnt and Entpd4 were significantly differentially expressed in six of the here analyzed tissues. On the other hand Gsr is significantly differentially expressed only in the small intestine tissues. We have clarified this in the revised manuscript.

5. It is unclear why the authors chose to focus on chylomicrons. Were additional molecules investigated but not revealed to be significant, or (perhaps more likely), is there a specific biological question being raised: if so, please make this question explicit.

In our study, we used the production of chylomicrons as an objective function for the small intestine since it is one of the major functions of the tissue. We also included HDL secretion since it is another major known function of the small intestine.

6. Please clarify your method for performing multiple hypotheses correction. The Benjamini-Hochberg method provides a False Discovery Rate, while Storey's method provides q-values. Which method (or methods) is used (or how they are combined) is unclear.

We used PIANO R package during the analysis of the gene expression data and The FDR (Benjamini Hochberg) method (Benjamini Hochberg, 1995) is the default P-value adjustment method in PIANO. In the revised version of the manuscript we rewrote as

"P-values were adjusted for multiple testing using the R-function p.adjust with the method set to false discovery rate (FDR) and Q-values were calculated. Q-values were used for network dependent analysis."

Additional minor comments:

1. The manuscript can be improved by a thorough proof-reading. Errors in grammar and spelling appear throughout.

Based on the suggestion of the reviewer, we asked our colleagues to read the manuscript and included their comments to the paper.

2. Please do not break clauses with parentheses. For example, consider "significantly higher (Q-value<0.05)" in place of "significantly (Q-value<0.05) higher".

We followed the suggestion of the reviewer and changed it throughout the manuscript.

Considering the above, we are confident that the revised version of our paper is of general interest to researchers in the field of metabolism, gut microbiota and systems biology.

Sincerely,

Adil Mardinoglu, PhD
Ass. Professor of Systems Medicine

Chalmers University of Technology & Royal Institute of Technology,
Email: adilm@chalmers.se & adilm@scilifelab.se
Tel: +46 (31) 772 31 40
Fax: +46 (31) 772 38 01

References

- Agren R, Mardinoglu A, Asplund A, Kampf C, Uhlen M, Nielsen J (2014) Identification of anticancer drugs for hepatocellular carcinoma through personalized genome-scale metabolic modeling. *Mol Syst Biol* **10**: 721
- Bordel S, Agren R, Nielsen J (2010) Sampling the solution space in genome-scale metabolic networks reveals transcriptional regulation in key enzymes. *Plos Comput Biol* **6**: e1000859
- Ghaffari P, Mardinoglu A, Asplund A, Shoaie S, Kampf C, Uhlen M, Nielsen J (2015) Identifying anti-growth factors for human cancer cell lines through genome-scale metabolic modeling. *Sci Rep* **5**: 8183
- Machado D, Herrgard M (2014) Systematic evaluation of methods for integration of transcriptomic data into constraint-based models of metabolism. *Plos Comput Biol* **10**: e1003580
- Uhlen M, Fagerberg L, Hallstrom BM, Lindskog C, Oksvold P, Mardinoglu A, Sivertsson A, Kampf C, Sjostedt E, Asplund A, Lundberg E, Djureinovic D, Odeberg J, Habuka M, Tahmasebpour S, Danielsson A, Edlund K, Szilagyarto CA, Skogs M, Takanen JO et al (2015) Tissue-based map of the human proteome. *Science* **347**: 1260419
- Uhlen M, Oksvold P, Fagerberg L, Lundberg E, Jonasson K, Forsberg M, Zwahlen M, Kampf C, Wester K, Hober S, Wernerus H, Bjorling L, Ponten F (2010) Towards a knowledge-based Human Protein Atlas. *Nat Biotechnol* **28**: 1248-1250